# Alignment Integration Network for Salient Object Detection and Its Application for Optical Remote Sensing Images

**DOI:** 10.3390/s23146562

**Published:** 2023-07-20

**Authors:** Xiaoning Zhang, Yi Yu, Yuqing Wang, Xiaolin Chen, Chenglong Wang

**Affiliations:** 1Changchun Institute of Optics, Fine Mechanics and Physics, Chinese Academy of Sciences, Changchun 130033, China; zhangxiaoning21@mails.ucas.ac.cn (X.Z.); wangyq@ciomp.ac.cn (Y.W.); chenxiaolin@ciomp.ac.cn (X.C.); wangchenglong@ciomp.ac.cn (C.W.); 2University of Chinese Academy of Sciences, Beijing 100049, China

**Keywords:** salient object detection, alignment integration, strip attention module, boundary enhancement module, optical remote sensing image

## Abstract

Salient object detection has made substantial progress due to the exploitation of multi-level convolutional features. The key point is how to combine these convolutional features effectively and efficiently. Due to the step by step down-sampling operations in almost all CNNs, multi-level features usually have different scales. Methods based on fully convolutional networks directly apply bilinear up-sampling to low-resolution deep features and then combine them with high-resolution shallow features by addition or concatenation, which neglects the compatibility of features, resulting in misalignment problems. In this paper, to solve the problem, we propose an alignment integration network (ALNet), which aligns adjacent level features progressively to generate powerful combinations. To capture long-range dependencies for high-level integrated features as well as maintain high computational efficiency, a strip attention module (SAM) is introduced into the alignment integration procedures. Benefiting from SAM, multi-level semantics can be selectively propagated to predict precise salient objects. Furthermore, although integrating multi-level convolutional features can alleviate the blur boundary problem to a certain extent, it is still unsatisfactory for the restoration of a real object boundary. Therefore, we design a simple but effective boundary enhancement module (BEM) to guide the network focus on boundaries and other error-prone parts. Based on BEM, an attention weighted loss is proposed to boost the network to generate sharper object boundaries. Experimental results on five benchmark datasets demonstrate that the proposed method can achieve state-of-the-art performance on salient object detection. Moreover, we extend the experiments on the remote sensing datasets, and the results further prove the universality and scalability of ALNet.

## 1. Introduction

As an important research branch in computer vision, salient object detection (SOD) has received much attention in recent years. It can serve as a fundamental pre-processing technique to facilitate various computer vision applications, such as foreground map evaluation [1], image retrieval [2], visual tracking [3,4], remote sensing image segmentation [5], and semantic segmentation [6].

Benefiting from the development of deep learning technology, great advancements [7,8,9,10,11,12,13,14,15,16,17,18,19,20,21,22,23,24,25,26,27,28,29,30,31,32,33,34,35,36,37] in SOD have been made. In [38], Wang et al. provide a comprehensive survey that reviews deep SOD algorithms from various aspects, including network architecture, level of supervision, and so on. As summarized by [38], most of the current deep learning based methods design their architectures based on fully convolutional networks (FCN) [39] to integrate multi-level convolutional features. However, due to stepwise down-sampling operations, features from different levels have contradictions, and the contextual information they possess is asymmetric, which results in *misalignment problems* during the feature aggregation process; current work tends to ignore this problem.

To address the misalignment problem, we explore various alignment technologies and propose a novel alignment integration network (ALNet) for SOD. Figure 1 illustrates the alignment processes of different technologies.

To increase interpretability of the models, we visualize the integrated feature maps of each model. An FCN-based model, which combines adjacent level features by direct addition, is utilized as the baseline model (i.e., w.o.Align). As we can see, features without alignment are fuzzy and unfocused. The important semantic and structural information is not well represented because of misalignment. Flow alignment (see Figure 1a), which has been proven to be effective in semantic segmentation [40], provides us with a feasible solution to alleviate the misalignment. Motivated by [40], we propose a flow alignment model to align adjacent level features for SOD. In flow alignment, semantic flow (i.e., offset Δ) is learned for spatial warping of high-level features. The visualized results in Figure 1 demonstrate the effectiveness of flow alignment. However, the flow alignment only learns one offset at each spatial position of a feature, which is sometimes not enough to handle complex misalignment. Therefore, we further propose a deformable alignment model (see Figure 1b) by substituting deformable convolution for spatial warping to increase the offset diversity for better alignment. Compared with flow alignment, deformable alignment can better highlight the salient region as well as maintain useful spatial details. The details of flow alignment and deformable alignment are explained in Section 3.2.

Moreover, the ability of a network to model global context is also critical to performance improvement. Recently, non-local self-attention mechanism [41] has been proven to be effective in capturing long-range dependencies. However, how to effectively incorporate it in SOD is still challenging. First of all, we need to consider computational efficiency. In this paper, we introduce strip attention [42] into our network to augment contextual information as well as ensure computational efficiency. Second, the adaptation of the self-attention mechanism for SOD is also an important factor to consider. Different from [42], where strip attention is utilized once to enhance the final feature for scene parsing, in our ALNet, strip attention modules (SAMs) are embedded in the intermediate procedure of alignment integration to augment contextual information for the high-level integrated features. Due to SAM, the global semantics are selectively incorporated in the alignment integration to recover precise salient objects.

Furthermore, to strengthen the model’s learning ability at the object boundary, we design a simple but effective boundary enhancement module, which can output an attention map for the network. Based on the attention map, an attention weighted loss (AW loss) function is proposed to make the network pay more attention to the ambiguous and hard regions. Features from this branch are utilized as a complement for the multi-level integrated features to conduct the final prediction.

Finally, to prove the robustness and scalability of the proposed method, we directly apply our network to optical remote sensing images (RSIs) and compare it with state-of-the-art RSI-SOD methods [32,43,44,45,46] (salient object detection methods that are specially designed for RSIs). The extensive experiments demonstrate the effectiveness of our method.

The main contributions of our proposed method are summarized as follows:We propose an alignment integration network (ALNet) to alleviate the misalignment problem in multi-level feature fusion, thereby generating effective representation for salient object detection.Strip attention is introduced into our network to augment global contextual information for the high-level integrated features as well as keep computational efficiency.To make the network focus more on the boundary and error-prone regions, we propose a boundary enhancement module and an attention weighted loss function to help the network generate results with sharper boundaries.Experimental results on SOD benchmarks as well as remote sensing datasets demonstrate the effectiveness and scalability of the proposed ALNet.

## 2. Related Work

Existing deep SOD methods can be roughly categorized into multi-level features integration based and boundary learning based approaches.

### 2.1. Integrating Multi-Level Features for SOD

A simple but effective way to integrate multi-level features is adding or concatenating features step by step, as with FCN [39], which is usually taken as a baseline model. However, in this direct integration way, associations between features cannot be well modeled, resulting in unsatisfactory performance. Compared with this direct way, Amulet [10] integrates multi-scale features in a fully connected way. Nevertheless, fusing features from all levels at every specific scale may introduce unnecessary redundant information. Based on FCN, PAGRN [12] introduces both channel-wise and spatial-wise attention to suppress the irrelevant interference from features and then combines attentive features by stepwise addition. Pyramid fusion structure is utilized by Wei et al. [23] to fuse high-level semantics with low-level details via lateral connections. In [17], Wang et al. design an ingenious network that conduct both top-down and bottom-up inference in an iterative and cooperative manner. The predicted saliency map is integrated with multi-level features step by step for coarse-to-fine saliency estimation. Sun et al. [28] leverage the average- and max-pooling modules to integrate the multi-level features in the spatial and channel-wise dimensions, respectively. An architecture search framework is proposed by Zhang et al. [29] to automatically learn a multi-scale features fusion strategy. All of the existing methods design ingenious modules to integrate features; nevertheless, they neglect the misalignment problem of multi-level features. To address this problem, we introduce alignment technology into SOD and further design an alignment integration network to relieve the misalignment for effective feature integration.

### 2.2. Boundary Learning for SOD

Precise salient object boundaries are beneficial for the performance of SOD methods. CNN-based methods suffer from blurred boundaries due to stride and pooling operations. Incorporating shallow layer features can alleviate the problem to a certain extent, but sometimes this is not enough. In order to obtain sharper object boundaries, some methods, such as [9,11,14], utilize CRF [47] as the post-processing step to enhance object edges. However, the post-processing operation is too time-consuming to be employed in real-time applications. In [16], Wang et al. design a salient edge detection module to emphasize the importance of boundary information, and L2-norm loss is employed to supervise salient edges. BASNet [20] employs a hybrid loss that incorporates SSIM [48] to capture the structural information in an image. Weighted BCE and IOU loss are utilized by F3Net [23], which synthesizes the local structure information of a pixel to guide the network to focus more on local details. In [29], Zhang et al. employ boundary loss [49] to penalize the misalignment of salient object boundaries. Mei et al. [37] adopt the patch-level edge preservation loss [50], which considers a local neighborhood of each pixel and assigns more attention to the object boundary. Different form these algorithms, in this paper, based on the boundary enhancement module, we propose an attention weighted loss, which can adaptively promote the network to focus on the hard pixels (i.e., pixels from boundaries or other error-prone parts).

## 3. Materials and Methods

In this section, we explain the details of our proposed ALNet, whose main framework is shown in Figure 2.

The backbone includes five convolutional blocks, which are ${\left\{\mathrm{Block}\ell \right\}}_{\ell =0}^{4}$. Multi-level features with different resolutions (i.e., 1/4, 1/8, 1/16, and 1/32 of the original resolution) are side-outputted from Block1 to Block4 and are denoted as ${\left\{X_{\ell }\right\}}_{\ell =1}^{4}$. Then, the features are sent to the pre-process module, which is explained in Section 3.1. Next, we propose the alignment integration module (AIM) to combine adjacent level features by feature alignment in Section 3.2. The boundary enhancement module (BEM), which is utilized to equip AIM to generate more powerful features, is explained in Section 3.3. Finally, we introduce the proposed attention weighted loss and the supervision strategy in our work in Section 3.4.

### 3.1. Pre-Process Module

As shown in Figure 2, shallower features ${\left\{X_{\ell }\right\}}_{\ell =1}^{3}$ are fed into 1×1 convolution followed by the batch norm and ReLU operations, respectively. As for the top-level feature (i.e., $X_{4}$), an additional 3×3 convolution is applied to extract high-level semantics for the network. After pre-processing, we can obtain multi-level features ${\left\{F_{\ell }\right\}}_{\ell =1}^{4}$. Then, alignment integration is carried out for them.

### 3.2. Alignment Integration Module

Most of the existing methods directly integrate multi-level features without considering the misalignment problem between them. To alleviate this problem, we propose a novel alignment integration module (AIM), which is constructed based on the feature alignment (FA). As shown in Figure 2, in AIM, adjacent level features conduct FA to generate an aligned feature, which is then fed into next FA with the shallower level feature, and so on. The procedures of FA are shown in Figure 1. For the adjacent level features $F_{\ell }$ and $F_{\ell +1}$, we first generate alignment offset for them.

#### 3.2.1. Offset Generation

First of all, $F_{\ell +1}$, which denotes a high-level feature with low resolution, is up-sampled to the same size as $F_{\ell }$. Next, we concatenate them together and take the concatenated features as the input for a 3×3 convolution layer to output the alignment offset:(1)$$\Delta_{\ell }=\mathrm{Conv}\left(\mathrm{Cat}\left(\mathrm{UP}\left(F_{\ell +1}\right),F_{\ell }\right)\right),$$
where Cat(·) and Up(·) denote the concatenation and bi-linear upsampling operation, respectively. Then, we conduct feature alignment for them.

#### 3.2.2. Feature Alignment

Two kinds of feature alignment models (i.e., flow alignment in Figure 1a and deformable alignment in Figure 1b), which intrinsically share the same formulation but differ in their offset diversity, are proposed in our work.

**Flow Alignment.** For flow alignment, the offset $\Delta_{\ell }\in R^{H_{\ell }\times W_{\ell }\times 2}$ is utilized for the spatial warping of $F_{\ell +1}$:(2)$${\tilde{F}}_{\ell +1}=T\left(F_{\ell +1},\Delta_{\ell }\right),$$
where T(·,·) represents the alignment transformation function; $\Delta_{\ell }$ consists of two feature maps, which represent the offset for x- and y-coordinates of each position on the feature map to be aligned, respectively. Let $T_{hw}$ denote the output of T(F,Δ). The function is defined as follows:(3)$$\begin{array}{c} T_{hw}=\sum_{h^{\prime }=1}^{H}\sum_{w^{\prime }=1}^{W}F_{h^{\prime }w^{\prime }}\cdot \max(0,1-|\frac{h+\Delta_{1hw}}{\delta }-h^{\prime }\left|\right)\cdot \max(0,1-|\frac{w+\Delta_{2hw}}{\delta }-w^{\prime }\left|\right), \end{array}$$
which samples features on position $p(\frac{h+\Delta_{1hw}}{\delta },\frac{w+\Delta_{2hw}}{\delta })$ of F and linearly interpolates the values of the four neighbors (top-left, top-right, bottom-left, and bottom-right) of p to approximate the output. The variable δ denotes the scale difference between F and Δ (e.g., when F’s resolution is half that of Δ, δ=2); $\Delta_{1hw}$ and $\Delta_{2hw}$ represent the learned 2D transformation offsets for position (*h*, *w*).

**Deformable Alignment.** As for deformable alignment, a 3×3 deformable convolution is utilized in our network. The number of offsets is in proportion to the kernel size of the deformable convolution. Therefore, the learned offset is $\Delta_{\ell }\in R^{H_{\ell }\times W_{\ell }\times 18}$. The feature is aligned by modulated deformable convolution (i.e., DCN-v2 [51]) based on the offset:(4)$${\tilde{F}}_{\ell +1}=\mathrm{DeformConv}\left(\mathrm{UP}\left(F_{\ell +1}\right),\Delta_{\ell }\right).$$

Let Y denote the output of DeformConv(F,Δ):(5)$$Y\left(p\right)=\sum_{k=1}^{n^{2}}\omega \left(p_{k}\right)\cdot F\left(p+p_{k}+\Delta_{p_{k}}\right)\cdot m_{k}\left(p\right),$$
where p is the spatial position, $p_{k}$ is the *k*th sampling offset in a standard convolution, and *n* is the kernel size of deformable convolution (i.e., 3); ω and m are learnable parameters in the DeformConv. Compared with flow alignment, deformable alignment adaptively learns the diverse offsets for features, thus can deal with the misalignment problem better, which corresponds with the experimental results in Section 5.1.

#### 3.2.3. Aligned Integration

The aligned integrated feature can be obtained by:(6)$${\tilde{F}}_{\ell }={\mathrm{CBR}}_{3\times 3}\left({\tilde{F}}_{\ell +1}+F_{\ell }\right),$$
where ${\mathrm{CBR}}_{3\times 3}\left(\cdot \right)$ denotes a 3×3 convolution with batch normalization and ReLU operations. In AIM, multi-level features are integrated step by step like FCN but alleviate misalignment. The integrated feature of the last step (i.e., ${\tilde{F}}_{1}$) is equipped with both semantic information and spatial details.

#### 3.2.4. Strip Attention Module

To augment the contextual information for the intermediate integrated features and promote their pixel-wise representative capacity, we incorporate non-local self-attention into our network. The standard non-local self-attention has a computational complexity of O((H×W)×(H×W)), where *H* and *W* denote the spatial dimensions of the input feature map. In this paper, we introduce strip attention [42], which reduces the computational complexity to O((H×W)×W) by a stripping operation to add global context as well as keep efficiency. The strip attention module (SAM) is displayed in Figure 3.

For simplicity, here we use $F\in R^{C\times H\times W}$ to denote the input feature. First, F is fed into three convolutional layers with 1×1 filters followed by batch normalization and ReLU to generate three new feature maps, which are $Q\in R^{C^{\prime }\times H\times W}$, $K\in R^{C^{\prime }\times H\times W}$, and $V\in R^{C\times H\times W}$, respectively; $C^{\prime }$ is an intermediate feature dimension number for variable Q and K. To make SAM efficient, we set $C^{\prime }$ smaller than *C*.

A stripping operation (i.e., average pooling with pooling windows of size H×1) is applied on K to encode global context representation in the vertical direction, and then we get $K\in R^{C^{\prime }\times 1\times W}$. We also try to apply 1×W pooling on the feature to incorporate context in the horizontal direction, but it has little effect on the performance improvement. Considering computation complexity, we only use a one direction stripping operation.

Next, we reshape Q and K to $R^{C^{\prime }\times N}$ and $R^{C^{\prime }\times W}$, respectively, where N=H×W. Then, we can calculate the strip attention map $\mathrm{SA}\in R^{N\times W}$ along the horizontal as follows:(7)$$\mathrm{SA}=softmax(Q^{T}★K),$$
where ★ means matrix multiplication and *T* means matrix transposition. Similarly, we apply stripping and reshape operations to V and can obtain $V\in R^{C\times W}$. Then, we conduct a matrix multiplication between SA and $V^{T}$ and reshape the result to get $F^{SA}\in R^{C\times H\times W}$. The output feature can be formulated as:(8)$$F^{\prime }=F+F^{SA}.$$

For inputs ${\tilde{F}}_{3}$ and ${\tilde{F}}_{2}$, the outputs of SAM are denoted as ${\tilde{F}}_{3}^{\prime }$ and ${\tilde{F}}_{2}^{\prime }$, respectively. As shown in Figure 2, after adding SAM in our network, when ℓ=1 or ℓ=2, input of Equations (Equation 2) and (Equation 4) should be ${\tilde{F}}_{\ell +1}^{\prime }$. For SOD, a high-level feature is expected to be augmented by global context, whereas a shallow-level feature is supposed to place emphasis on structural details. Therefore, we do not add SAM in the shallow level integration (i.e., level 1 in Figure 2). The experimental results in Section 5.2 demonstrate the rationality of our design (i.e., SAM-ver vs. SAM-ver-1).

### 3.3. Boundary Enhancement Module

An auxiliary boundary enhancement branch, which is simple but effective, is proposed to guide the network focus on boundaries and other error-prone parts of the image. The boundary enhancement module (BEM) is illustrated in Figure 4.

We apply two convolution followed by batch normalization operations on the input feature to generate attention map *A*, which is utilized as a weight for the loss computation in Section 3.4. Ground-truth boundary maps, which are pre-computed by the method in [52], are used to provide guidance for the attention generation. In addition, we extract the intermediate feature as guidance to enhance and complement the input feature. As shown in Figure 4, the input feature and the guidance are concatenated together and fused by a 1×1 convolution with batch normalization and ReLU operations. The enhanced feature is then used to conduct salient object prediction.

### 3.4. Supervision Strategy

In this paper, a hybrid loss function is proposed to supervise the network. At first, we introduce BCE [53] and IOU loss [54] to ensure pixel-wise smooth gradient as well as optimize the global structure. For the saliency map *S* and ground truth G, the BCE loss can be calculated as follows:(9)$$\begin{array}{c} L_{B}\left(S,G\right)=-\sum_{x=1}^{H}\sum_{y=1}^{W}\left[G_{xy}\log\left(S_{xy}\right)+\left(1-G_{xy}\right)\log\left(1-S_{xy}\right)\right], \end{array}$$
where (x,y) denotes the spatial position; *H* and *W* represent the height and width of images. IOU loss is formulated as:(10)$$L_{I}\left(S,G\right)=1-\frac{\sum_{x=1}^{H}\sum_{y=1}^{W}S_{xy}G_{xy}}{\sum_{x=1}^{H}\sum_{y=1}^{W}\left[S_{xy}+G_{xy}-S_{xy}G_{xy}\right]}.$$

Furthermore, to boost the network to learn sharper boundaries, we propose an *attention weighted loss* (AW loss) based on the learned attention map *A* in Section 3.3. The AW loss can be considered as a combination of attention weighted BCE and IOU loss:(11)$$\begin{array}{c} L_{AW}\left(S,G,A\right)=L_{AWB}\left(S,G\right)+L_{AWI}\left(S,G\right)\;\;\;\;\;\;\;\;\;\;\;\;\;\;\;\;\;\;\;\;\;\\ =-\frac{\sum_{x=1}^{H}\sum_{y=1}^{W}\left[G_{xy}\log\left(S_{xy}\right)+\left(1-G_{xy}\right)\log\left(1-S_{xy}\right)\right]A_{xy}}{\sum_{x=1}^{H}\sum_{y=1}^{W}A_{xy}}\\ +1-\frac{\sum_{x=1}^{H}\sum_{y=1}^{W}S_{xy}G_{xy}A_{xy}}{\sum_{x=1}^{H}\sum_{y=1}^{W}\left[S_{xy}+G_{xy}-S_{xy}G_{xy}\right]A_{xy}}.\;\;\;\;\; \end{array}$$

In addition, to ensure the attention map focuses on the boundary, we use an auxiliary weighted BCE Loss $L_{AX}\left(A,G_{b}\right)$ to supervise *A*, where $G_{b}$ is the ground-truth boundary (radius = 2) generated from *G*. The calculation of $L_{AX}$ is as in [55].

The final loss function for the proposed network is as follows:(12)$$L=L_{B}+L_{I}+\beta L_{AW}+\lambda L_{AX},$$
where β=1 and λ=20 are weighting coefficients for the loss function. We set the parameters based on experimental experience.

## 4. Results

Experimental results of the proposed work are displayed in this section. In Section 4.1 and Section 4.2, we introduce the datasets and evaluation metrics of the experimental results. Implementation details of the proposed ALNet are described in Section 4.3. In Section 4.4, we compare our method with the state-of-the-art models from both quantitative and qualitative aspects. Furthermore, we conduct extension experiments on optical remote sensing images (RSIs) and compare our ALNet with state-of-the-art RSI-SOD methods. The details are introduced in Section 4.5.

### 4.1. Datasets

The experiments are conducted on five benchmark datasets: ECSSD [56], HKU-IS [57], PASCAL-S [58], DUT-OMRON [59], and DUTS [60]. The ECCSD dataset contains 1000 natural images with complex structures. In HKU-IS, there are 4447 images, which include multiple salient objects or objects touching the image boundary. PASCAL-S, which is generated from the PASCAL VOC dataset [61], contains 850 images. DUT-OMRON is a challenging dataset with 5168 images. DUTS is a relatively large dataset that contains 10,553 training images and 5019 testing images. We train our network based on the training images of DUTS for salient object detection.

In addition, in order to further demonstrate the stability and scalability of ALNet, we test the proposed method on two optical remote sensing datasets dedicated to SOD: ORSSD [44] and EORSSD [43]. ORSSD is the first publicly available dataset for SOD in optical remote sensing images. It contains 800 images (600 for training and 200 for testing), which are collected from the Google Earth and some existing RSI datasets. EORSSD is a large public dataset for RSI-SOD that extends ORSSD to 2000 images (1400 for training and 600 for testing). Specifically, we augment the training set of EORSSD and ORSSD by flipping and rotation, generating seven additional variants of the original training data. On EORSSD, we train our ALNet based on 11,200 augmented pairs. On ORSSD, we train our ALNet with 4800 augment pairs.

### 4.2. Metrics

We adopt the popular precision–recall (PR) curves, F-measure curves, mean F-measure ($F_{\beta }$) [62], weighted F-measure ($F_{\beta }^{\omega }$) [63], mean absolute error (*M*) [64], and mean E-measure ($E_{\xi }^{m}$) [1] as our evaluation metrics. Mean F-measure is an overall performance measurement, which is defined as:(13)$$F_{\beta }=\frac{(1+\beta^{2})\times Precision\times Recall}{\beta^{2}\times Precision+Recall},$$
where $\beta^{2}=0.3$ to emphasize the precision. Weighted F-measure offers an intuitive generalization of mean F-measure by changing precision and recall to their ωth power. As suggested in [65], $\beta^{2}$ for the weighted F-measure is set to 1.0. Mean absolute error is defined as the average pixel-wise absolute difference between the binary ground truth *G* and the saliency map *S*, which can be computed by:(14)$$MAE=\frac{1}{W\times H}\sum_{x=1}^{W}\sum_{y=1}^{H}\left|S\left(x,y\right)-G\left(x,y\right)\right|,$$
where *W* and *H* denote width and height of saliency map, respectively. The E-measure focuses on both local pixel values and image-level statistics. It can be computed by:(15)$$E_{\xi }=\frac{1}{W\times H}\sum_{x=1}^{W}\sum_{y=1}^{H}\theta \left(\xi \right),$$
where θ(ξ) is the enhanced alignment matrix. Mean E-measure ($E_{\xi }^{m}$) is utilized in our experiment.

### 4.3. Implementation Details

The proposed method is based on the Pytorch platform. We conduct our experiments on a PC with an Intel Core i7-9700KF CPU (with 3.9 GHz Turbo boost) and a single NVIDIA GTX 2080Ti GPU. The input images are resized to 352 × 352 for both training and testing. We use data augmentation methods such as normalizing, cropping, and flipping. The parameters of the backbone are initialized from VGG16 [66], ResNet50 [67], and MSCAN-b [68] for fair comparison with existing methods. We utilize SGD optimizer [69] to train the entire network end to end. The base learning rate is set to 0.05, and the warm-up and linear decay strategies are used to adjust the learning rate. The momentum and the weight decay are set to 0.9 and 1 × 10^−4^, respectively. Batch size is set to 30 (for ResNet50 backbone) and 20 (for VGG16 and MSCAN-b backbone), and we train the network for 60 epochs. Apex (https://github.com/NVIDIA/apex (accessed on 20 December 2022)) and fp16 are utilized to accelerate the training process.

For extended experiments on remote sensing datasets, the implementation details are just the same as the original SOD. The only difference is the training dataset. Specially, on the EORSSD and ORSSD datasets, we resize the input image to 288 × 288 and train our ALNet for 65 epochs and 45 epochs, respectively.

For VGG16, ResNet50, and MSCAN-b backbone, the inference time of the proposed method for a 352 × 352 image is 0.0235 s (43 fps), 0.0188 s (53 fps), and 0.0280 s (36 fps), respectively, which demonstrates the feasibility of our method for real-time applications. The source code will be released to facilitate reproducibility.

### 4.4. Comparison to State-of-the-Art Methods

We compare our proposed algorithm with 17 state-of-the-art salient object detection methods, including AFNet [15], PAGENet [16], PS [17], ASNet [18], CPD [19], BASNet [20], EGNet [21], SCRN [22], F3Net [23], GateNet [24], GCPANet [25], ITSD [26], MINet [27], A-MSF [29], VST [35], ICON [36], and DCENet [37].

For fair comparisons, we directly use the saliency maps offered by the authors or use the provided codes to generate the results. As some algorithms employ various backbones, we compare with the best results of them.

**Quantitative Comparison.** Table 1 shows the quantitative comparison results in terms of mean F-measure, weighted F-measure, mean absolute error, and mean E-measure. We also compare the computational complexity and the size of parameters in the second and third columns of Table 1 (i.e., MACs and Params). Figure 5 shows the P-R curves and F-measure curves of our method and the state-of-the-art methods. From the results, we can see that, for VGG16 and ResNet50 backbone, our proposed network performs favorably against other state-of-the-art methods on all datasets and metrics, as well as keeps the complexity and model size relatively small, which demonstrates the effectiveness of our proposed network based on alignment integration.

For the attention based backbone, we implement our network based on MSCAN-b, which utilizes multi-scale convolution attention to encode features. Compared with the existing methods, ALNet-MS ranks first on most of the datasets and metrics. It is noteworthy that ALNet-MS has smaller MACs and Params than the existing methods. The experiments based on different backbones all prove that our proposed network can achieve state-of-the-art performance in both effectiveness and efficiency.

**Qualitative Comparison.** In Figure 6, we compare the visual results of the methods for qualitative evaluation. Benefiting from multi-level alignment integration, our network can generate powerful integrated features, which contain both high-level semantics and spatial details, to segment salient regions even in very challenging scenes (e.g., 1st and 2nd rows in Figure 6). In addition, compared with other boundary learning based methods such as F3Net and A-MSF, our proposed methods can generate relatively clear and accurate object boundaries.

### 4.5. Extension Experiment on the Remote-Sensing Datasets

To further discuss the proposed model’s robustness and scalability, we conduct experiments on optical remote sensing datasets. We compare our ALNet with four state-of-the-art RSI-SOD methods: LVNet [44], DAFNet [43], MJRB [45], and ACCoNet [46]. For fair comparison with existing methods, the network is initialized from ResNet50.

**Quantitative Comparison.** The quantitative comparison results of mean F-measure, weighted F-measure, mean absolute error, and mean E-measure are shown in Table 2. For the EORSSD dataset, the proposed method ranks first on all metrics. For the ORSSD dataset, the result of our method is also competitive. In Figure 7, we display the F-measure curves of the proposed method with state-of-the-art methods on two remote sensing datasets. Our method performs well against state-of-the-art RSI-SOD methods. It is worth mentioning that the proposed method is a universal framework for salient object detection and not dedicated to optical remote sensing images. However, the results demonstrate the effectiveness and scalability of the proposed network.

**Qualitative Comparison.** The qualitative results, including several challenging and representative scenes of optical remote sensing images, are shown in Figure 8.

For the first scene (i.e., object with shadows), being affected by the shadows, ACCoNet, MJRB, DAFNet, and LVNet cannot generate accurate and sharp boundaries, but our method can better highlight the object and produce relatively accurate results.

For the scene with a tiny object, which is typical in optical remote sensing images, our proposed method can segment the tiny object with fine details; compared with the other methods, the object shape generated by our method is closer to the ground truth.

Another difficult scene is one with multiple objects. As shown in Figure 8, ACCoNet and MJRB incorrectly predict non-salient interference in the background as foreground. DAFNet generates blur salient regions, and LVNet fails to detect the real objects in the first row of this scene. In contrast, our method captures all objects finely without any redundant regions.

For the scene with irregular geometry structure (e.g., lakes and rivers), the saliency maps of our method obviously have sharper boundaries, and the highlighted regions are concentrated. From the visual results, we can see that our methods can better deal with the complex and challenging scenes in optical remote sensing images, which further proves the reliability of the method.

## 5. Discussion

In this section, we conduct ablation studies for all the proposed modules (i.e., feature alignment, strip attention module, and boundary enhancement module) in our ALNet and analyze the effectiveness of them in Section 5.1, Section 5.2, and Section 5.3, respectively. For a comprehensive analysis of the model, we further discuss the failure cases in Section 5.4.

### 5.1. Effectiveness of Feature Alignment

We use an FCN-based model, which combines adjacent level features by direct addition, as the baseline model (i.e., w.o.Align). Different alignment technologies are exploited on the baseline model. The results based on ResNet50 are shown in the first part of Table 3. F-Align and and D-Align denote flow alignment and deformable alignment, respectively. The comparisons of the alignment methods and the baseline demonstrate the misalignment problem in multi-level feature integration indeed decreases the performance. Deformable alignment performs better than flow alignment, which indicates the importance of offset diversity.

In addition, we visualize the last-stage integrated features of different methods to make the results explainable, as shown in Figure 9. As we can see, features without alignment are fuzzy and lack both semantic and structural information. After alignment, the models can generate more meaningful feature representation. Compared with flow alignment, deformable alignment features can better highlight salient regions and have more precise boundaries, which coincides with the quantitative results in Table 3.

### 5.2. Effectiveness of SAM

On the basis of D-Align, we conduct stripping operations in both the vertical and horizontal directions for the integrated feature. In the second part of Table 3, we list the results of using vertical stripping (+SAM-ver), using horizontal stripping (+SAM-hori), and using both directions (+BSAM). The experimental results demonstrate SAM is effective for adaptively encoding global contextual relations for the integrated feature. SAM-ver is superior to SAM-hori on most of the datasets, and using both directions did not bring out improvement. On the basis of SAM-ver, we add SAM on the integration of feature level 1 (SAM-ver-1). The results show that SAM-ver is better than SAM-ver-1, which indicates shallow level integration prefers spatial details to global context.

Furthermore, SAM is essentially a simplified self attention mechanism, and to further prove its effectiveness, we compare it with a non-local module in Table 3. Compared with non-local, SAM performs better in our network, and, due to stripping operations, SAM is more efficient in computation.

### 5.3. Effectiveness of BEM

The boundary enhancement module is a simple but effective branch for the network to generate clear boundaries. In the third part of Table 3, we conduct ablation studies for BEM based on +SAM-ver. The results indicate that BEM is effective for performance improvement. Removing $L_{AX}$ or $L_{AW}$ lowers the final results, which proves the effectiveness of each part in BEM.

In Figure 10, we compare the saliency maps with and without BEM and visualize attention map *A* at the same time.

From the results, we can see that BEM can learn reasonable attention maps, which make the network put more emphasis on boundary and error-prone regions. The results with BEM obviously have sharper boundaries and can deal with more complex backgrounds.

### 5.4. Failure Cases

The failure cases of our method are displayed in Figure 11.

In the first row, our method incorrectly predicts oranges as the foreground objects. In the second row, the whole bed (not just the pillow) is taken as the salient object by our method. In the third row, our method cannot detect the real object (i.e., the board with “organic”). Similarly, other state-of-the-art methods also fail in these cases. We summarize the possible reasons for these failure cases: (1) insufficient training samples (e.g., 1st row); (2) controversial annotations (e.g., 2nd row); (3) too complex scene and requirements for additional information like depth (e.g., 3rd row).

## 6. Conclusions

In this paper, an alignment integration network (ALNet) is proposed to alleviate misalignment problems in combining multi-level convolutional features. Feature alignment is designed in our network to align adjacent level features step by step to produce effective feature representation for salient object detection. To help the network encode global context, a strip attention module is introduced to augment the representative capacity of the feature. Finally, we construct a boundary enhancement module and an attention weighted loss function to make the network focus on boundaries and hard regions. Comprehensive experiments are conducted on five SOD benchmarks and two remote sensing datasets. The experimental results demonstrate the state-of-the-art performance of our ALNet as well as the effectiveness of each proposed modules.

## Figures and Tables

**Figure 1 sensors-23-06562-f001:**
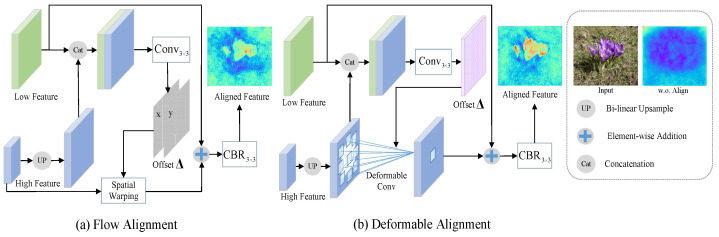
Illustration of various alignment technologies. ${\mathrm{CBR}}_{3\times 3}$ means a 3×3 convolution followed by batch normalization and ReLU operations. ${\mathrm{Conv}}_{3\times 3}$ is a 3×3 convolution to generate offset. The feature maps of each model are visualized by averaging along the channel dimension. Larger values are denoted by hot colors, and vice versa.

**Figure 2 sensors-23-06562-f002:**
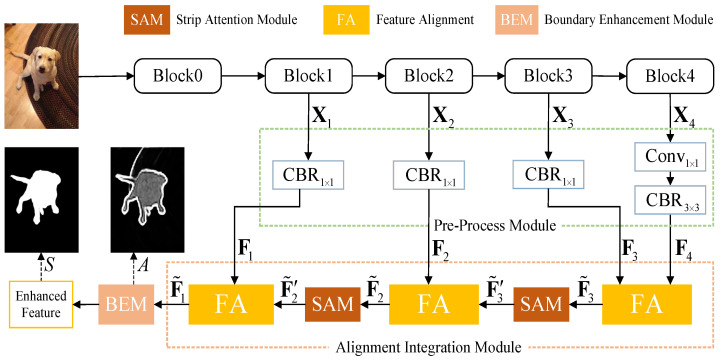
Main framework of our alignment integration network; ${\mathrm{CBR}}_{k\times k}$ means a k×k convolution followed by batch normalization and ReLU operations. We first side-out the multi-level convolutional features from the backbone and process them by the pre-process module. An additional 3×3 convolution operation is applied on the top level feature to encode high-level semantics. Then, features from multi-level are fed into the alignment integration module, in which adjacent level features are progressively combined by feature alignment. A strip attention module is utilized to capture non-local contextual information for the intermediate integrated feature. The final integrated feature is further enhanced by a boundary enhancement module, and the enhanced feature is exploited to conduct salient object prediction.

**Figure 3 sensors-23-06562-f003:**
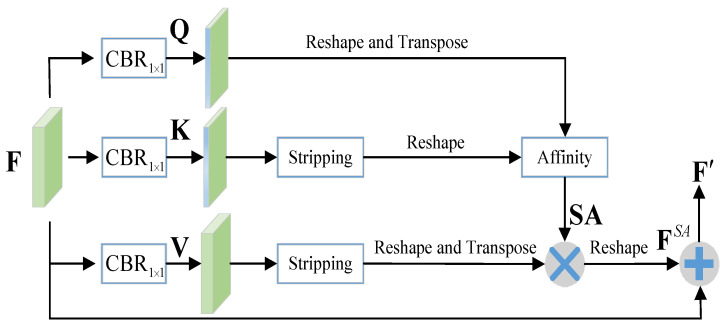
Illustration of the strip attention module. Stripping operations are utilized to reduce the computational complexity of this module.

**Figure 4 sensors-23-06562-f004:**
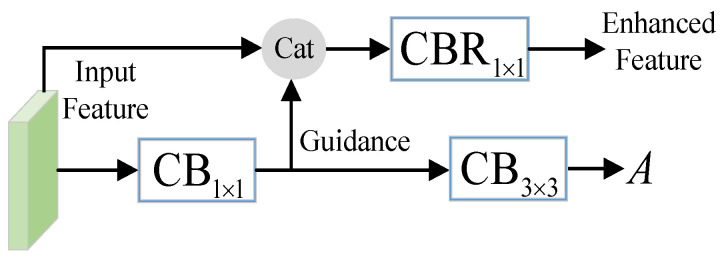
Illustration of boundary enhancement module; ${\mathrm{CB}}_{k\times k}$ means a k×k convolution followed by batch normalization operation. The input feature is processed to generate an attention map for attention weight loss computation, and the intermediate feature is taken as a guidance to produce the enhanced feature.

**Figure 5 sensors-23-06562-f005:**
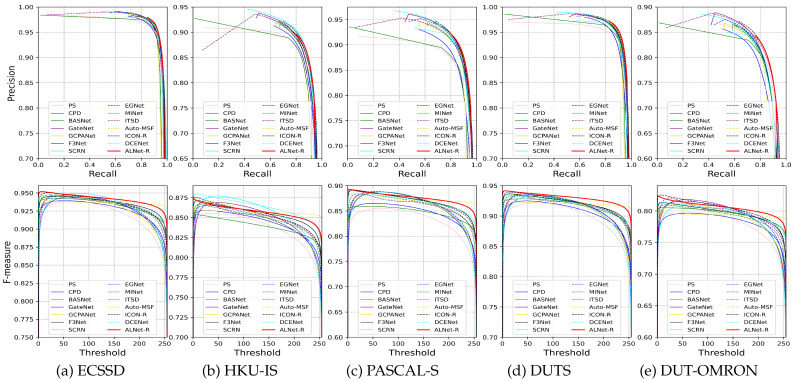
P-R curves and F-measure curves of the proposed method compared with other state-of-the-art methods on five benchmark datasets.

**Figure 6 sensors-23-06562-f006:**
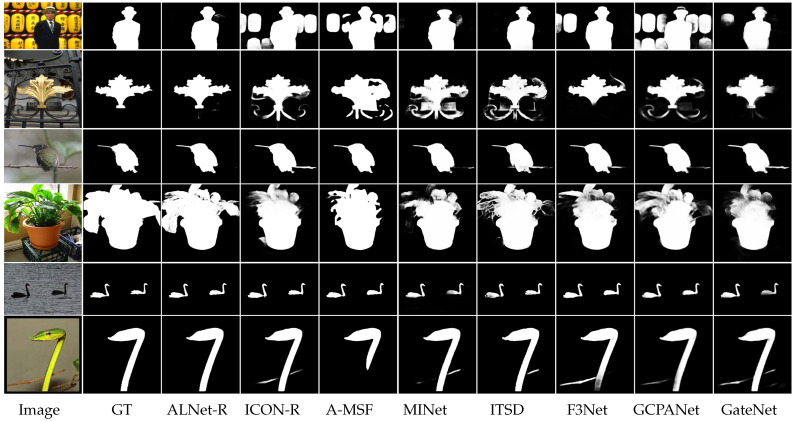
Visual comparisons of our results and the state-of-the-art methods. Our method can uniformly highlight salient regions and produce sharper boundaries even with complex background distractions in the scene.

**Figure 7 sensors-23-06562-f007:**
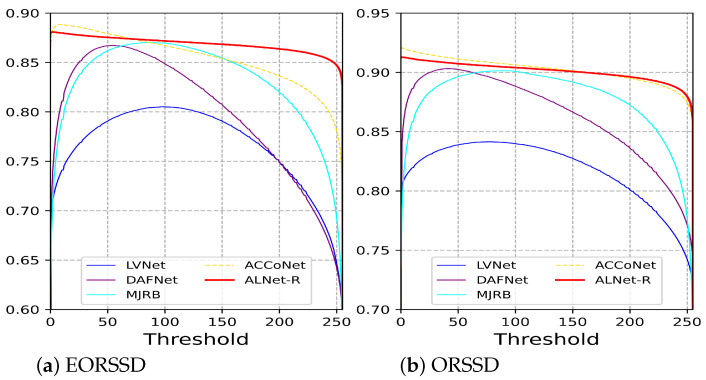
F-measure curves of the proposed method with state-of-the-art RSI-SOD methods on two remote sensing datasets.

**Figure 8 sensors-23-06562-f008:**
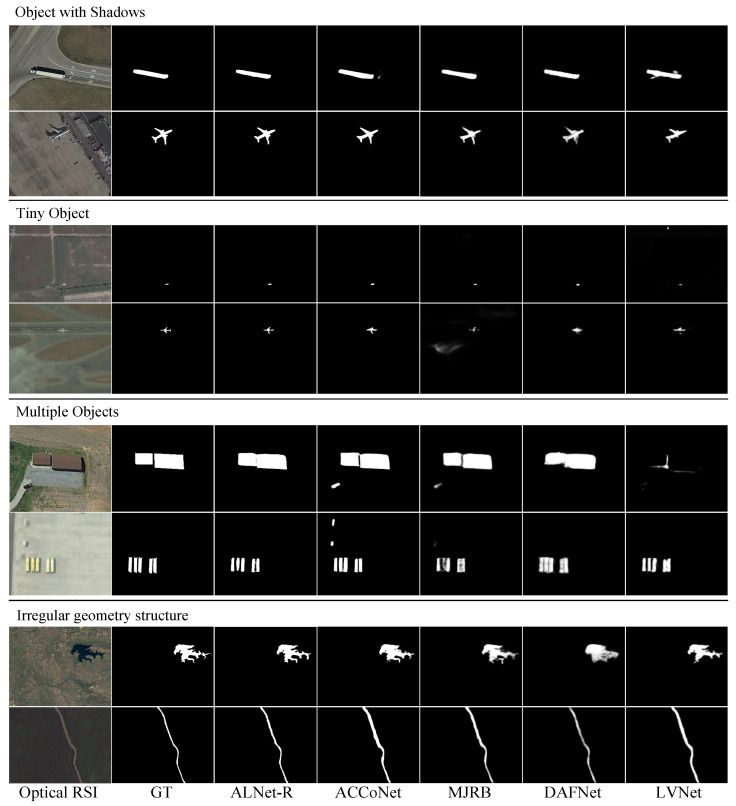
Visual comparisons with four representative state-of-the-art RSI-SOD methods.

**Figure 9 sensors-23-06562-f009:**
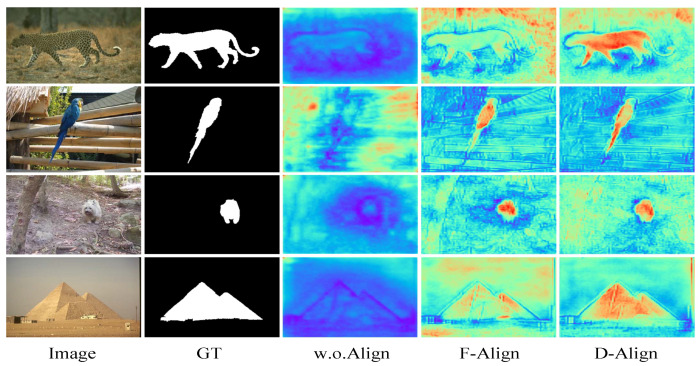
Visualization of feature maps with and without alignment. Larger values are denoted by hot colors, and vice versa.

**Figure 10 sensors-23-06562-f010:**
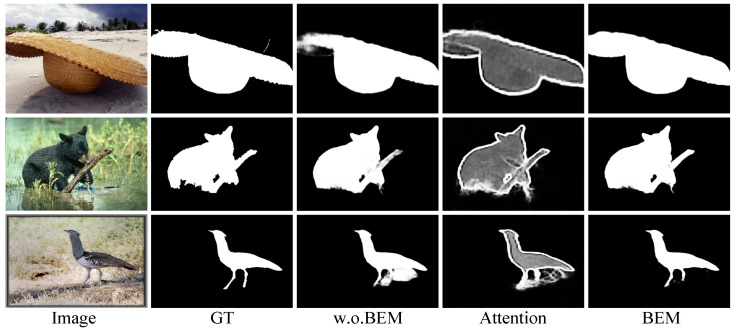
Visual comparisons for BEM. Attention map *A* generated by BEM is also visualized; w.o.BEM means without BEM in our network.

**Figure 11 sensors-23-06562-f011:**
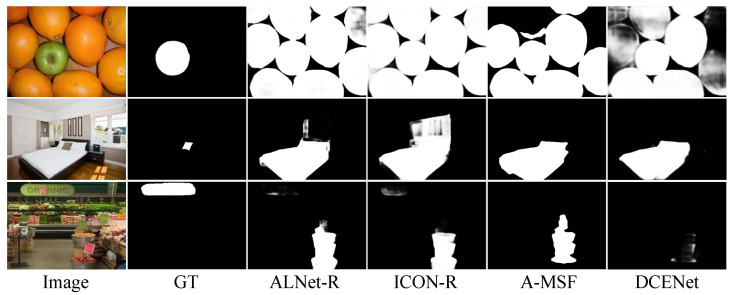
Failure cases of our proposed method and other state-of-the-art methods.

**Table 1 sensors-23-06562-t001:** Comparisons with 17 methods on 5 benchmark datasets. The best two results of each part are shown in red and blue; ↑ means higher value is better, whereas ↓ is the contrary. ‘-V’: VGG16 [66], ‘-R’: ResNet50 [67], ‘-T2’: T2T-ViT [70], ‘-S’: SWIN [71], ‘-MS’: MSCAN-b [68].

Method	MACs	Params	ECSSD				HKU-IS				PASCAL-S				DUTS				DUT-OMRON			
			$F_{\beta }↑$	$F_{\beta }^{\omega }↑$	M↓	$E_{\xi }^{m}↑$	$F_{\beta }↑$	$F_{\beta }^{\omega }↑$	M↓	$E_{\xi }^{m}↑$	$F_{\beta }↑$	$F_{\beta }^{\omega }↑$	M↓	$E_{\xi }^{m}↑$	$F_{\beta }↑$	$F_{\beta }^{\omega }↑$	M↓	$E_{\xi }^{m}↑$	$F_{\beta }↑$	$F_{\beta }^{\omega }↑$	M↓	$E_{\xi }^{m}↑$
VGG based Backbone																						
PAGENet (2019)	–	–	0.904	0.886	0.042	0.936	0.884	0.865	0.037	0.935	0.811	0.783	0.076	0.878	0.793	0.769	0.052	0.883	0.743	0.722	0.062	0.849
AFNet (2019)	21.66	35.95	0.905	0.886	0.042	0.935	0.888	0.869	0.036	0.934	0.824	0.797	0.070	0.883	0.812	0.785	0.046	0.893	–	–	–	–
ASNet (2020)	–	–	0.890	0.865	0.047	0.926	0.873	0.846	0.041	0.923	0.817	0.784	0.070	0.882	0.760	0.715	0.061	0.854	–	–	–	–
ALNet-V	48.24	15.95	0.928	0.915	0.033	0.950	0.920	0.910	0.027	0.956	0.836	0.815	0.064	0.900	0.853	0.836	0.037	0.921	0.765	0.744	0.056	0.864
ResNet50 based Backbone																						
PS (2019)	–	–	0.904	0.881	0.041	0.937	0.883	0.856	0.038	0.933	0.814	0.780	0.071	0.883	0.804	0.762	0.048	0.892	0.760	0.730	0.061	0.867
CPD (2019)	17.7	47.85	0.913	0.898	0.037	0.942	0.892	0.875	0.034	0.938	0.819	0.794	0.071	0.882	0.821	0.795	0.043	0.898	0.742	0.719	0.056	0.847
BASNet (2019)	127.36	87.06	0.917	0.904	0.037	0.943	0.902	0.889	0.032	0.943	0.818	0.793	0.076	0.879	0.822	0.803	0.048	0.895	0.767	0.751	0.056	0.865
EGNet (2019)	157.21	111.69	0.918	0.903	0.037	0.943	0.902	0.887	0.031	0.944	0.823	0.795	0.074	0.881	0.839	0.815	0.039	0.907	0.760	0.738	0.053	0.857
SCRN (2019)	15.09	25.23	0.916	0.900	0.037	0.939	0.894	0.876	0.034	0.935	0.833	0.807	0.063	0.892	0.833	0.803	0.040	0.900	0.749	0.720	0.056	0.848
F3Net (2020)	16.43	25.54	0.924	0.912	0.033	0.948	0.910	0.900	0.028	0.952	0.835	0.816	0.061	0.898	0.851	0.835	0.035	0.920	0.766	0.747	0.053	0.864
GateNet (2020)	162.13	128.63	0.913	0.894	0.040	0.936	0.897	0.880	0.033	0.937	0.826	0.797	0.067	0.886	0.837	0.809	0.040	0.906	0.757	0.729	0.055	0.855
GCPANet (2020)	54.31	67.06	0.916	0.903	0.035	0.944	0.901	0.889	0.031	0.944	0.829	0.808	0.062	0.895	0.841	0.821	0.038	0.911	0.756	0.734	0.056	0.853
ITSD (2020)	15.96	26.47	0.921	0.910	0.034	0.947	0.904	0.894	0.031	0.947	0.831	0.812	0.066	0.894	0.840	0.823	0.041	0.913	0.768	0.750	0.061	0.865
MINet (2020)	87.11	126.38	0.923	0.911	0.033	0.950	0.909	0.897	0.029	0.952	0.830	0.809	0.064	0.896	0.844	0.825	0.037	0.917	0.757	0.738	0.056	0.860
A-MSF (2021)	17.5	32.5	0.927	0.916	0.033	0.951	0.912	0.903	0.027	0.956	0.842	0.822	0.061	0.901	0.855	0.841	0.034	0.928	0.772	0.757	0.050	0.873
DCENet (2022)	59.78	192.96	0.924	0.913	0.035	0.948	0.908	0.898	0.029	0.951	0.845	0.825	0.061	0.902	0.849	0.833	0.038	0.918	0.769	0.753	0.055	0.865
ICON-R (2023)	20.91	33.09	0.928	0.918	0.032	0.954	0.912	0.902	0.029	0.953	0.838	0.818	0.064	0.899	0.853	0.836	0.037	0.924	0.779	0.761	0.057	0.876
ALNet-R	19.82	28.46	0.932	0.923	0.030	0.955	0.921	0.913	0.026	0.959	0.843	0.826	0.059	0.907	0.860	0.847	0.035	0.928	0.778	0.761	0.055	0.874
Attention based Backbone																						
VST-T2 (2021)	23.16	44.63	0.920	0.910	0.033	0.951	0.907	0.897	0.029	0.952	0.835	0.816	0.061	0.902	0.845	0.828	0.037	0.919	0.774	0.755	0.058	0.871
ICON-S (2023)	52.59	94.30	0.940	0.936	0.023	0.966	0.929	0.925	0.022	0.968	0.865	0.854	0.048	0.924	0.893	0.886	0.025	0.954	0.815	0.804	0.043	0.900
ALNet-MS	15.14	27.45	0.943	0.938	0.024	0.964	0.936	0.932	0.020	0.969	0.866	0.851	0.051	0.922	0.899	0.893	0.024	0.955	0.817	0.806	0.043	0.903

**Table 2 sensors-23-06562-t002:** Comparisons with four state-of-the-art RSI-SOD methods on two remote sensing datasets. The best two results are shown in red and blue.

Methods	Backbone	EORSSD				ORSSD			
		$F_{\beta }↑$	$F_{\beta }^{\omega }↑$	M↓	$E_{\xi }^{m}↑$	$F_{\beta }↑$	$F_{\beta }^{\omega }↑$	M↓	$E_{\xi }^{m}↑$
LVNet (2019)	VGG	0.736	0.702	0.015	0.882	0.800	0.775	0.021	0.926
DAFNet (2021)	ResNet	0.784	0.783	0.006	0.929	0.851	0.844	0.011	0.954
MJRB (2022)	ResNet	0.806	0.792	0.010	0.921	0.857	0.842	0.015	0.939
ACCoNet (2023)	ResNet	0.846	0.852	0.007	0.966	0.895	0.896	0.009	0.977
ALNet-R	ResNet	0.865	0.865	0.006	0.967	0.895	0.892	0.009	0.975

**Table 3 sensors-23-06562-t003:** Ablation study of our proposed method. We show the results based on the ResNet50 backbone. The table can be divided into three parts to demonstrate effectiveness of the proposed modules in ALNet. Best results are shown in **bold**.

Settings	ECSSD			PASCAL-S			DUTS			DUT-OMRON		
	$F_{\beta }↑$	$F_{\beta }^{\omega }↑$	M↓	$F_{\beta }↑$	$F_{\beta }^{\omega }↑$	M↓	$F_{\beta }↑$	$F_{\beta }^{\omega }↑$	M↓	$F_{\beta }↑$	$F_{\beta }^{\omega }↑$	M↓
Effectiveness of Alignment												
w.o.Align	0.898	0.883	0.043	0.814	0.790	0.070	0.808	0.787	0.045	0.727	0.701	0.063
F-Align	0.921	0.908	0.035	0.834	0.812	0.062	0.847	0.830	0.038	0.757	0.736	0.056
D-Align	0.923	0.910	0.034	0.837	0.816	0.062	0.852	0.835	0.037	0.766	0.747	0.056
Effectiveness of SAM												
+SAM-ver	0.928	0.917	0.032	0.838	0.818	0.063	0.859	0.845	**0.035**	0.777	0.758	0.054
+SAM-hori	0.924	0.912	0.033	0.841	0.822	0.061	0.856	0.842	0.036	0.773	0.755	0.056
+BSAM	0.927	0.916	0.032	0.837	0.818	0.064	0.854	0.839	0.037	0.768	0.749	0.062
+SAM-ver-1	0.926	0.915	0.033	0.842	0.822	0.062	0.855	0.840	0.036	0.769	0.750	0.058
+Non-Local	0.924	0.912	0.034	0.837	0.815	0.064	0.852	0.835	0.037	0.772	0.752	0.055
Effectiveness of BEM												
+BEM	**0.932**	**0.923**	**0.030**	**0.843**	**0.826**	**0.059**	**0.860**	**0.847**	**0.035**	**0.778**	**0.761**	0.055
w/o $L_{AX}$	0.927	0.917	0.031	0.836	0.817	0.064	0.855	0.841	0.036	0.771	0.753	0.057
w/o $L_{AW}$	0.928	0.917	0.031	**0.843**	0.824	0.061	0.855	0.841	**0.035**	0.777	0.759	**0.053**

## Data Availability

The source code and datasets will be available at https://github.com/zhangxiaoning666/ (acessed on 18 July 2023).

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
