# Peer review of "Alignment Integration Network for Salient Object Detection and Its Application for Optical Remote Sensing Images"

_sensors, 2023, doi:10.3390/s23146562_

Round 1
Reviewer 1 Report
1. AI is already a proper term for artificial intelligence, so it is better to change the AI in AINet to ALNet. All alignment integration in this article can be represented by AL abbreviation, because the focus is on alignment rather than artificial intelligence.
2. All OURS in the text are changed to ALNet.
3. Do you use tensorflow? Use model.save to access the .h5 file, and then list the complete network architecture diagram. How many neurons are needed for each layer? How many epochs are calculated? What is the learning rate? What optimizer is used?
4. Please write down the full name of FCN.
5. Why is there no need for dropout and dense layers? How to correct weighting in theory? How to generate the .hdf5 file generated by the experimental results?
Reviewer 2 Report
The paper introduces the Alignment Integration Network (AINet) to address the misalignment issue in multi-level convolutional feature combination. The key contributions, as highlighted, include the Strip Attention Module (SAM), the Boundary Enhancement Module (BEM), and an attention-weighted loss function. However, this paper could be improved considering the following points:
1.The introduction could benefit from being more compact. Avoid complex terminologies or in-depth details in this section, which could be saved for the method and results sections.
2. The authors should also provide a discussion on the time complexity and the cost of the proposed method.
3. What is the choice of hyperparameters β and λ in the final loss function? Were the parameters based on previous research or heuristic methods?
There are some typos and grammar issues that should carefully checked.
Round 2
Reviewer 1 Report
All my concerned comments have been explained and revised.